# A RISK-SENSITIVE POLICY GRADIENT METHOD

## ABSTRACT

Standard deep reinforcement learning (DRL) agents aim to maximize expected reward, considering collected experiences equally in formulating a policy. This differs from human decision-making, where gains and losses are valued differently and outlying outcomes are given increased consideration. It also wastes an opportunity for the agent to modulate behavior based on distributional context. Several approaches to distributional DRL have been investigated, with one popular strategy being to evaluate the projected distribution of returns for possible actions. We propose a more direct approach, whereby the distribution of full-episode outcomes is optimized to maximize a chosen function of its cumulative distribution function (CDF). This technique allows for outcomes to be weighed based on relative quality, does not require modification of the reward function to modulate agent behavior, and may be used for both continuous and discrete action spaces. We show how to achieve an asymptotically consistent estimate of the policy gradient for a broad class of CDF-based objectives via sampling, subsequently incorporating variance reduction measures to facilitate effective on-policy learning. We use the resulting algorithm to train agents with different "risk profiles" in penalty-based formulations of six OpenAI Safety Gym environments, observing that moderate emphasis on improvement in training scenarios where the agent performs poorly both increases the accumulation of positive rewards and decreases the frequency of incurred penalties. We found that, in all environments tested, the same risk profile can be used to produce both stronger overall performance than standard Proximal Policy Optimization (PPO) and higher levels of positive reward than PPO constrained by Lagrangians to maintain the same cost levels.

## 1 INTRODUCTION

While deep reinforcement learning (DRL) has been used to master an impressive array of simulated tasks in controlled settings, it has not yet been widely adopted for high-stakes, real-world applications. One reason for this gap is the lack of distributional perspective in standard artificial agents. Endowing agents with such perspective could make their decision-making more robust, potentially leading to increased safety, increased trust from humans, and more widespread real-world adoption.

In reinforcement learning (RL), *risk* arises due to uncertainty around the possible outcomes of an agent's future actions. It is a result of randomness in the operating environment, mismatch between training and test conditions, and the inherent randomness of a stochastic policy. *Risk-sensitive* policies, or those that consider more than a mean over the distribution of possible outcomes, offer the potential for added robustness under uncertain and dynamic conditions. There is an evolving landscape of algorithmic paradigms for handling risk in RL, from constraint-based approaches adapted from optimal control (Achiam et al., 2017; Chow et al., 2019; Ray et al., 2019; Zhong et al., 2020) to adversarial approaches emerging from AI Safety (García & Fernández, 2015; Amodei et al., 2016). Within this landscape, learning approaches that optimize distributional measures offer the ability to express design preferences over the full distribution of potential outcomes, through the specification of a risk-sensitivity criterion.

Distributional RL has been studied for value-based methods, with a popular strategy being to use the distributional Bellman equation to estimate the distribution of Q-values for each member of a discrete set of potential actions (Bellemare et al., 2017; Dabney et al., 2018a;b). However, distributional RL has not been widely explored for policy gradient methods, which could permit direct optimization of risk-sensitive measures and naturally accommodate both discrete and continuous action spaces.

In the following, we introduce a novel framework for risk-sensitive learning using policy gradients. Our approach allows agents to be trained with different risk profiles through design-time specification of both utility and weight functions, with the latter being defined over the estimated distribution of full-episode rewards. This framework enables agent-based learning that captures aspects of human decision-making, such as overemphasis of rare occurrences and diminishing marginal utility relative to a reference outcome (Kahneman & Tversky, 1979). It also allows implementation of another key strategy of human learning: emphasizing improvement on tasks where one is deficient. We demonstrate the ability of our algorithm to use this strategy to improve performance relative to both unconstrained and constrained methods in six OpenAI Safety Gym environments (Ray et al., 2019).

## 2 RELATED WORK

Constrained RL offers a set of approaches to safe exploration (García & Fernández (2015)) that aim to enforce explicit constraints throughout the learning process via methods including Lagrangian constraints (Ray et al. (2019)) and constraint coefficients (Achiam et al. (2017)). Differing from the safe exploration scenario, we here consider problems with distinct training and test phases where agent performance is to be evaluated. Our experiments indicate that risk-sensitive learning can offer performance improvements over constrained learning in scenarios where safety constraints need not be enforced during training.

Distributional approaches to risk-sensitive RL have primarily been explored in the value-based setting. Therein, the value distribution has been explicitly modeled through categorical techniques (Bellemare et al., 2017) or quantile regression (Dabney et al., 2018a) and used to improve both value predictions and overall performance. Recent works utilize distributional modeling in the actor-critic setting to enable application to continuous action spaces, again demonstrating improved performance over baseline approaches (Ma et al., 2020; Zhang et al., 2021; Duan et al., 2021). In value-based approaches, risk-sensitivity criteria are applied at run time as a nonlinear warping of the estimated value distribution.

Policy gradient approaches offer additional promise for risk-sensitive RL, but require direct optimization of a parameterized policy with respect to a distributional objective. Some existing methods are limited to a specific class of learning objective, such as the set of concave risk measures that allow a globally-optimal solution (Tamar et al., 2015; Zhong et al., 2020). Others allow a broader class of measures but are more restrictive in the class of policies that can be represented (Prashanth et al., 2016). We aim for a risk-sensitive policy gradient approach that both offers significant flexibility in the choice of learning objective and can learn policies parameterized by a deep neural network.

Various measures have been considered in the context of risk-sensitive RL, including exponential utility (Pratt, 1964), percentile performance criteria (Wu & Lin, 1999), value-at-risk (Leavens, 1945), conditional value-at-risk (Rockafellar & Uryasev, 2000), and prospect theory (Kahneman & Tversky, 1979). In this work, we consider a class of risk-sensitivity measures motivated by Cumulative Prospect Theory (CPT) (Tversky & Kahneman, 1992). CPT uniquely models two key aspects of human decision-making: (1) a utility function $u$, computed relative to a reference point that induces more risk-averse behavior in the presence of gains than losses and (2) a weight function $w$ that prioritizes outlying events. Specific forms of $u$ and $w$ are given in Tversky & Kahneman (1992) (and in Appendix A.4), but the general form of CPT admits a wide variety of risk-sensitive objectives.

Here we show how to train agents to optimize this class of objectives through sampling-based estimation of their policy gradients and requisite variance reduction. The final algorithm resembles well-known on-policy approaches such as Proximal Policy Optimization (Schulman et al., 2017b) and is similarly widely applicable. Although we do not explore it here, the incorporation of an appropriate risk-sensitivity criterion could additionally enable risk-aware exploration and adversarial training for increased robustness (Pinto et al., 2017; Parisi et al., 2019; Zhang et al., 2020).

## 3 RISK-SENSITIVE POLICY OPTIMIZATION

In this section we formalize the class of distributional objectives to be considered, derive a sampling-based approximation of its policy gradient, enact variance reduction on this estimate, and use the result to produce a practical learning algorithm.

### 3.1 Preliminaries: Problem and Notation

Standard deep reinforcement learning seeks to maximize the expected reward of an agent over encountered trajectories; that is, it maximizes the objective

$$J(\theta) = E_{\tau \sim p_\theta(\tau)} \left[ \sum_t r(\mathbf{s}_t, \mathbf{a}_t) \right]. \tag{1}$$

Here $p_\theta(\tau)$ is the distribution over trajectories $\tau \equiv \mathbf{s}_1, \mathbf{a}_1, \ldots, \mathbf{s}_T, \mathbf{a}_T$ induced by a policy parameterized by $\theta$; $\mathbf{s}_t$, $\mathbf{a}_t$, and $r(\mathbf{s}_t, \mathbf{a}_t)$ denote the state, action, and reward at time $t$, respectively. To enable the incorporation of distributional context, we instead consider the objective

$$J(\theta) = \int_{-\infty}^{+\infty} u(r(\tau)) \frac{d}{dr(\tau)} \left( w(P_\theta(r(\tau))) \right) dr(\tau), \tag{2}$$

where $u(r(\tau))$ is the utility associated with full-trajectory reward $r(\tau) \equiv \sum_t r(\mathbf{s}_t, \mathbf{a}_t)$ and $w$ is a piecewise differentiable weighting function of the CDF of trajectory reward $P_\theta(r(\tau)) = \int_{-\infty}^{r(\tau)} p_\theta(r') dr'$.

Equation 2 is inspired by CPT (Tversky & Kahneman, 1992), which includes a pair of integrals of this form. It was chosen for its generality; by using different utility functions $u$ and weight functions $w$ one may represent all of the risk measures mentioned in Section 2 and all of the risk measures evaluated by Dabney et al. (2018a). The form (2) reduces to (1) when $u$ and $w$ are both the identity mapping. While designed for the episodic setting, the objective (2) may be considered for infinite horizons through the choice of appropriately long windows.

### 3.2 Risk-Sensitive Policy Gradient

To optimize the objective (2), we first derive an approximation to its gradient with respect to the policy parameters $\theta$. Working toward a representation that can be sampled, we assert the independence of the reward on $\theta$ and use the chain rule to write

$$\nabla_\theta J(\theta) = \int_{-\infty}^{\infty} u(r(\tau)) \frac{d}{dr(\tau)} \left( w'(P_\theta(r(\tau))) \nabla_\theta P_\theta(r(\tau)) \right) dr(\tau), \tag{3}$$

where $w'$ is the derivative of $w$ with respect to $P_\theta(r(\tau))$. The gradient of the CDF may be written as follows:

$$\nabla_\theta P_\theta(r(\tau)) = \nabla_\theta \int_{-\infty}^{r(\tau)} p_\theta(r') dr' = \nabla_\theta \int_{\tau'} H(r(\tau) - r(\tau')) p_\theta(\tau') d\tau'$$
$$= \int_{\tau'} H(r(\tau) - r(\tau')) \nabla_\theta p_\theta(\tau') d\tau' = \int_{\tau'} H(r(\tau) - r(\tau')) p_\theta(\tau') \nabla_\theta \log p_\theta(\tau') d\tau'. \tag{4}$$

Here the first equality is the integral representation of $P_\theta(r(\tau))$, the second uses the Heaviside step function to select all trajectories with total reward $\leq r(\tau)$, the third follows from the independence of reward on $\theta$, and the fourth follows from the expression for the derivative of the natural logarithm. In the following, we also use the complementary expression

$$\nabla_\theta P_\theta(r(\tau)) = \nabla_\theta \left( 1 - \int_{r(\tau)}^{\infty} p_\theta(r') dr' \right) = - \int_{\tau'} H(r(\tau') - r(\tau)) p_\theta(\tau') \nabla_\theta \log p_\theta(\tau') d\tau'. \tag{5}$$

Either form, or a combination of the two, may be substituted into (3) and the result sampled over $N$ trajectories by first ordering trajectories $i = 1 \ldots N$ by increasing reward $r(\tau)$. Then

$$\nabla_\theta J(\theta) \approx \sum_{i=1}^{N} u(r(\tau_i)) \left( w'\left(\frac{i}{N}\right) \nabla_\theta P_\theta(r(\tau_i)) - w'\left(\frac{i-1}{N}\right) \nabla_\theta P_\theta(r(\tau_{i-1})) \right), \tag{6}$$

where the term $w'(0)\nabla_\theta P_\theta(r(\tau_0)) \equiv 0$. This ordering scheme produces an asymptotically consistent estimate, as shown in the context of CPT value estimation by Prashanth et al. (2016). $\nabla_\theta P_\theta(r(\tau_i))$ may be sampled in one of two ways, based on either (4) or (5):

$$\nabla_\theta P_\theta(r(\tau_i)) \approx \frac{1}{N}\sum_{j=1}^{i}\sum_{t=1}^{T_j}\nabla_\theta\log\pi_\theta(\mathbf{a}_{j,t}|\mathbf{s}_{j,t}) \approx -\frac{1}{N}\sum_{j=i+1}^{N}\sum_{t=1}^{T_j}\nabla_\theta\log\pi_\theta(\mathbf{a}_{j,t}|\mathbf{s}_{j,t}). \quad (7)$$

The expression (6) may be used to train a policy that optimizes the distributional objective (2) in a manner similar to REINFORCE (Williams, 1992).

### 3.3 VARIANCE REDUCTION

Reducing the variance of sample-based gradient estimates enables faster learning. Here we take several steps to reduce the variance of (6), as has been done with the policy gradient estimate of REINFORCE (Williams, 1992).

First, note that cross-trajectory terms of the form $f(\tau_i, \mathbf{a}_{j,t}, \mathbf{s}_{j,t}) = u(r(\tau_i))\nabla_\theta\log\pi_\theta(\mathbf{a}_{j,t}|\mathbf{s}_{j,t})$, while nonzero, do not contribute to the gradient estimate in expectation when $i \neq j$. A proof of this assertion is given in Appendix A.1. Using (4) for the first term of (6) and (5) for the second allows us to write

$$\nabla_\theta J(\theta) \approx \sum_{i=1}^{N} u(r(\tau_i))\bigg(w'\bigg(\frac{i}{N}\bigg)\frac{1}{N}\sum_{j=1}^{i}\sum_{t=1}^{T_j}\nabla_\theta\log\pi_\theta(\mathbf{a}_{j,t}|\mathbf{s}_{j,t})$$
$$+ w'\bigg(\frac{i-1}{N}\bigg)\frac{1}{N}\sum_{j=i}^{N}\sum_{t=1}^{T_j}\nabla_\theta\log\pi_\theta(\mathbf{a}_{j,t}|\mathbf{s}_{j,t})\bigg). \quad (8)$$

Removing cross-trajectory terms gives

$$\nabla_\theta J(\theta) \approx \frac{1}{N}\sum_{i=1}^{N} u(r(\tau_i))\bigg(\bigg(w'\bigg(\frac{i}{N}\bigg) + w'\bigg(\frac{i-1}{N}\bigg)\bigg)\bigg)\sum_{t=1}^{T_i}\nabla_\theta\log\pi_\theta(\mathbf{a}_{i,t}|\mathbf{s}_{i,t}). \quad (9)$$

Note that the weight coefficients $(w'(\frac{i}{N}) + w'(\frac{i-1}{N}))$ should be normalized over each batch. The expression (9) is equivalent to (6) in expectation, but with reduced variance (see Appendix A.1 for justification). It has a clear intuition – trajectories are assigned utilities based on their rewards and their contributions to the gradient are scaled by the derivative of the weight function, just as they are in Cumulative Prospect Theory (Tversky & Kahneman, 1992).

Standard variance reduction techniques may be applied to this simplified form. Without further assumption or introduction of additional bias, a static baseline $b$ can be employed:

$$\nabla_\theta J(\theta) \approx \frac{1}{N}\sum_{i=1}^{N}\bigg(u(r(\tau_i)) - b\bigg)\bigg(w'\bigg(\frac{i}{N}\bigg) + w'\bigg(\frac{i-1}{N}\bigg)\bigg)\sum_{t=1}^{T_i}\nabla_\theta\log\pi_\theta(\mathbf{a}_{i,t}|\mathbf{s}_{i,t}) \quad (10)$$

Justification for this assertion is given in Appendix A.2. Learning may be further improved if we additionally assume that utility may be allocated on a per-step basis. In this case, per-step utilities are computed as the difference between what the full-episode utility would be if the episode were to end at a given time step and what it would have been had the episode ended at the previous time step. While not applicable in cases where episode utility is adjusted based on final outcome, this assumption has the significant benefit of modeling the temporal allocation of rewards and aligns with the standard formulation of RL. With it, the variance of (9) may be further reduced through the incorporation of utility-to-go and a state-dependent baseline $V_\phi(\mathbf{s}_{i,t})$:

$$\nabla_\theta J(\theta) \approx \frac{1}{N}\sum_{i=1}^{N}\bigg[w'\bigg(\frac{i}{N}\bigg) + w'\bigg(\frac{i-1}{N}\bigg)\bigg]\sum_{t=1}^{T_i}\nabla_\theta\log\pi_\theta(\mathbf{a}_{i,t}|\mathbf{s}_{i,t})\bigg[\sum_{t'=t}^{T_i}u(\mathbf{s}_{i,t'},\mathbf{a}_{i,t'}) - V_\phi(\mathbf{s}_{i,t})\bigg] \quad (11)$$

Here $u(\mathbf{s}_{i,t'},\mathbf{a}_{i,t'})$ is the per-step utility. The value function $V_\phi(\mathbf{s}_{i,t})$ is parameterized by $\phi$ and trained via regression to minimize

$$\mathcal{L}(\phi) = \sum_{i,t}\bigg(V_\phi(\mathbf{s}_{i,t}) - \sum_{t'=t}^{T_i}u(\mathbf{s}_{i,t'},\mathbf{a}_{i,t'})\bigg)^2. \quad (12)$$

A standard argument, similar to the approach taken in (Achiam, 2018), can be used to show that the incorporation of utility-to-go does not change the expected value of (9). The addition of a state-dependent baseline also does not introduce additional bias, as justified in Appendix A.2.

Finally, discount factors, bootstrapping, and trust regions may be used to provide additional variance reduction, just as they are in conventional on-policy learning (Appendix A.2). These measures may introduce additional bias to the policy gradient estimate, but typically lead to more sample-efficient learning. In our experiments, we evaluate the use of generalized advantage estimation (GAE; (Schulman et al., 2016)) based on the utility-to-go as well as the clipping-based trust regions of Proximal Policy Optimization (PPO; (Schulman et al., 2017b)). Incorporating these in the policy gradient yields

$$\nabla_\theta J(\theta) \approx \frac{1}{N} \sum_{i=1}^{N} \left( w'\left(\frac{i}{N}\right) + w'\left(\frac{i-1}{N}\right) \right) \sum_{t=1}^{T_i} \nabla_\theta L_{\text{clip}}\left( \log \pi_\theta(\mathbf{a}_{i,t}|\mathbf{s}_{i,t}), A_u^\pi(\mathbf{s}_{i,t}, \mathbf{a}_{i,t}) \right), \quad (13)$$

where $A_u^\pi(\mathbf{s}_{i,t}, \mathbf{a}_{i,t})$ is the standard GAE except with per-step utilities in place of rewards. Trust regions are implemented similarly to PPO, pessimistically clipping policy updates to be within a multiplicative factor of $1 \pm \epsilon$ of the existing policy:

$$\begin{aligned}
L_{\text{clip}} = \min \Bigg( & \log \pi_\theta(\mathbf{a}_{i,t}|\mathbf{s}_{i,t}) A_u^\pi(\mathbf{s}_{i,t}, \mathbf{a}_{i,t}), \\
& \log\left( \text{clip}\left( \frac{\pi_\theta(\mathbf{a}_{i,t}|\mathbf{s}_{i,t})}{\pi_{\theta_{\text{old}}}(\mathbf{a}_{i,t}|\mathbf{s}_{i,t})}, 1 \pm \epsilon \right) \pi_{\theta_{\text{old}}}(\mathbf{a}_{i,t}|\mathbf{s}_{i,t}) \right) A_u^\pi(\mathbf{s}_{i,t}, \mathbf{a}_{i,t}) \Bigg).
\end{aligned} \quad (14)$$

They are used to perform multiple policy updates using the same batch of data, providing learning that is no longer strictly on-policy but that can be significantly more sample efficient. When following this route, we apply the same early stopping mechanism, based on the Kullback-Leibler divergence ($D_{\text{KL}}$) between old and new policies, as was used by Ray et al. (2019).

### 3.4 LEARNING ALGORITHM

The above sample-based estimate of the policy gradient may be used to train agents to maximize distributional objectives of the form (2). The resulting method, Cumulative Prospect Proximal Policy Optimization (C3PO), is given in Algorithm 1 and mirrors standard on-policy learning.

---

**Algorithm 1** Cumulative Prospect Proximal Policy Optimization (C3PO)

**Require:** Policy: initial parameters $\theta_0$, learning rate $\alpha_\theta$, updates per batch $M_\theta$
**Require:** Value: initial parameters $\phi_0$, learning rate $\alpha_\phi$, updates per batch $M_\phi$
**Require:** Early stopping threshold $D_{\text{KL, stop}}$, discount factor $\gamma$
  **for** $k = 1, 2, \ldots$ **do**
    Collect set of episodes $\mathcal{D}_k = \{\tau_i\}$ by running policy $\pi(\theta_k)$ in the environment
    Compute per-step utilities $u(\mathbf{s}_{i,t}, \mathbf{a}_{i,t})$
    Fit value function by regression:
    **for** $m = 1, \ldots M_\phi$ **do**

$$\phi \leftarrow \phi + \alpha_\phi \nabla_\phi \frac{1}{\sum_i T_i} \sum_{i,t} \left( V_\phi(\mathbf{s}_{i,t}) - \sum_{t'=t}^{T_i} \gamma^{t'-t} u(\mathbf{s}_{i,t'}, \mathbf{a}_{i,t'}) \right)^2$$

    **end for**
    Update utility-based advantage estimates $A_u^\pi(\mathbf{s}, \mathbf{a})$, using new $V_\phi(\mathbf{s})$
    Compute weight coefficients based on ordered episode outcomes and normalize
    Update policy, using KL-based early stopping:
    **for** $m = 1, \ldots M_\theta$ **do**
      **if** $D_{\text{KL}}(\pi_\theta || \pi_{\theta_{\text{old}}}) < D_{\text{KL, stop}}$ **then**
        $\theta \leftarrow \theta + \alpha_\theta \frac{1}{N} \sum_{i=1}^{N} (w'(\frac{i}{N}) + w'(\frac{i-1}{N})) \sum_{t=1}^{T_i} \nabla_\theta L_{\text{clip}}(\log \pi_\theta(\mathbf{a}_{i,t}|\mathbf{s}_{i,t}), A_u^\pi(\mathbf{s}_{i,t}, \mathbf{a}_{i,t}))$
      **else**
        break
      **end if**
    **end for**
  **end for**

---

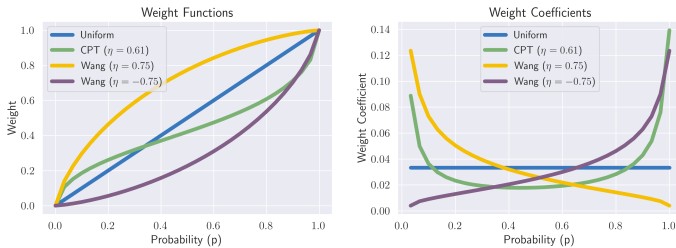

Figure 1: Example weight functions and their resulting coefficients in the policy gradient estimate (9). In these plots, outcomes increase in quality from left to right. As in CPT (Tversky & Kahneman, 1992), weight coefficients are proportional to the derivative of the weight function.

Beyond the utility and weight components, Algorithm 1 differs from conventional methods in the requirement to collect full episodes of data in each batch. This requirement can be removed if outcomes can be defined over partial rather than full episodes, an assumption that is often viable and matches human decision-making. For instance, while out of scope for this work, our approach could be applied to the Atari suite (Bellemare et al., 2013) by considering the outcomes of fixed-length windows and restructuring Algorithm 1 to mimic minibatch PPO (Schulman et al., 2017b).

## 4 EXPERIMENTS

To evaluate our approach, we sought to both establish that it can effectively optimize different distributional objectives and explore the impact of using different objectives on agent outcomes. We found the OpenAI Safety Gym (Ray et al., 2019) to be suitable for these purposes. Safety Gym is a configurable suite of continuous, multidimensional control tasks wherein different types of robots must navigate through obstacles with different dynamics to perform different tasks. By including both positive and negative events in each training scenario, it allowed us to evaluate how our various agents handled risk. Safety Gym is also highly stochastic: the locations of the goals and obstacles are randomized, leading to outcome variability and forcing the agent to learn a generalized navigation strategy.

Safety Gym logs adverse events but does not incorporate them into the reward function. As our method relies solely on the training signal from the reward, we assigned each logged adverse event a fixed, negative reward contribution in experiments using it or other unconstrained agents. Our initial experiments were conducted with a reward contribution of $-0.025$, which was found to allow agents to prioritize reaching goals but deter them from collisions with obstacles. To further emphasize obstacle avoidance, we doubled this contribution to $-0.05$ in our experiments using cautious weightings. These choices and the role they play are further discussed in Section 5.

To highlight distributional differences, we focused on the publicly available, obstacle-rich level 2 environments.[1] Avoiding the longer compute time of the "Doggo" robot, we evaluated the "Point" and "Car" robots on each task ("Goal", "Button", and "Push"). Further details on these environments and our rationale for choosing them are given in Appendix A.3.

In all experiments, we evaluated five random seeds and matched the hyperparameters used in the baselines accompanying Safety Gym (Ray et al., 2019) as closely as possible. The neural networks used to model both policy and value were multilayer perceptrons (MLPs), with two hidden layers of 256 units each and $\tanh$ activations. As in Ray et al. (2019), the policy network outputs the mean values of a multivariate gaussian with diagonal covariance. The control variances are optimized but independent of state. The full complement of variance reduction measures were used throughout; see Appendix A.4 for experimental justification of this choice.

### 4.1 DIFFERING OBJECTIVES

Agent performance was explored under four different distributional objectives. In addition to expected reward and CPT (configured to match the original form of Tversky & Kahneman (1992) and as given in Appendix A.4), we optimized for cautious ($\eta = 0.75$) and aggressive ($\eta = -0.75$) versions of the

---

[1]In Safety Gym, the default environments have three levels (0, 1, 2); obstacle density increases with level.

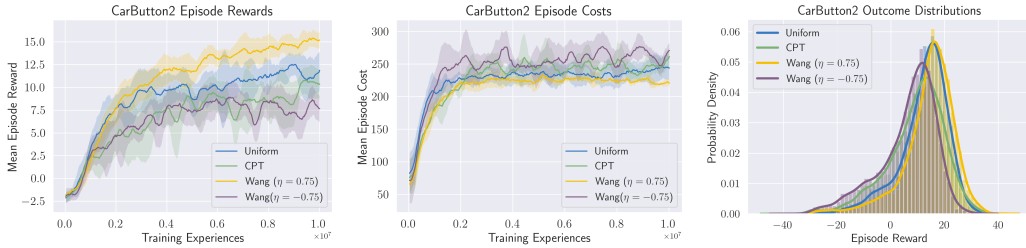

Figure 2: Impact of different distributional objectives in one environment (CarButton2). The shading in the first 2 plots (and subsequent learning curves) reflects the standard deviation associated with running over 5 random seeds. Left: Net reward (positive reward minus penalty) throughout learning. Middle: Average number of cost events per episode during training (lower is better). Right: Agent outcome distribution in testing (with sampling turned off). The cautious (Wang ($\eta = 0.75$)) weighting shows higher reward and lower cost once trained.

.

distortion risk measure proposed in Wang (2000). This measure is defined as $w(p) = \Phi(\Phi^{-1}(p) + \eta)$, where $\Phi$ and $\Phi^{-1}$ are the standard normal cumulative distribution function and its inverse. While we found this form to be convenient, the "Pow" metric in Dabney et al. (2018a) or any other set of similarly shaped $w$ curves should achieve a similar effect. In experiments using the objective from Tversky & Kahneman (1992), the reference point was taken to be the mean episode reward of the current batch, matching the tendency of humans to change their standards over time. The four weight functions and their resulting coefficients in (9) are shown in Figure 1.

Plots of the total rewards (including penalties) in training, average cost events per episode in training, and outcome distributions in testing are shown for one environment in Figure 2 and for two additional environments in Appendix A.5. The trends were fairly consistent over the three environments evaluated in this manner. While no explicit effort was made to handle cost (agents were given only the sum of positive rewards and penalties), the cautious and aggressive weightings consistently accumulated relatively low and high costs, respectively. The cautious (Wang($\eta = 0.75$)) agent typically also generated the highest positive and total rewards after 10 millions steps of training.

To generate the histograms in Figures 2, 5, 8, 9, and 13 as well as the numbers in Table 1, the trained agents were deployed on a set of 5000 test episodes – 1000 for each of the 5 networks learned using different random seeds in training. The resulting distributions therefore include contributions from both aleatoric and epistemic uncertainty. Sampling was turned off, allowing the agents to choose their perceived optimal action at each time step. In this context the benefit of emphasizing the lower part of the outcome distribution (i.e., cautious weighting) became more pronounced, in part because the methods that emphasize poor outcomes tended to maintain higher policy entropy (Appendix A.5).

## 4.2 CAUTIOUS WEIGHTINGS

To further explore the apparent benefits of cautious weightings in Safety Gym, we trained a series of variably cautious agents by tuning $\eta$ in the risk-averse weight function proposed by Wang (2000). Histograms of their episode rewards in testing are given in Appendix A.5 and summarized in Table 1. In these environments, agent performance – both in terms of improving the lower end of the reward distribution and on average – was seen to generally improve with increasing $\eta$ until the range $\eta \in [0.75, 1.25]$, subsequently degrading. Additional comparisons were made with PPO (Schulman et al., 2017b), which unsurprisingly was found to closely track performance of the "Uniform" agent. We found that naively incorporating cautious weightings into PPO improved its performance (row PPO + Wang(0.75) in Table 1), though not to the level of the full C3PO method with $\eta = 0.75$.

We then pursued a set of longer runs to compare C3PO with the cautious objective from Wang (2000) to both unconstrained and constrained benchmarks. Here we did not tune $\eta$, keeping it fixed at 0.75 for all experiments. Comparisons with unconstrained methods for three environments are given in Figures 3, 4, and 5 and for the remaining three environments in Appendix A.6. In addition to PPO, we compared performance with Trust Region Policy Optimization (TRPO; Schulman et al. (2017a)) as configured in Ray et al. (2019). Since this TRPO configuration generally outperformed the PPO configuration in Ray et al. (2019) from which we derived the hyperparameters for C3PO, we would expect C3PO to be at a disadvantage compared to TRPO. However, we found C3PO had the highest

|  | **PointButton2** | | | | **CarGoal2** | | | | **CarButton2** | | | |
|---|---|---|---|---|---|---|---|---|---|---|---|---|
|  | Mean | Std | Q=0.5 | Q=.05 | Mean | Std | Q=0.5 | Q=.05 | Mean | Std | Q=0.5 | Q=.05 |
| **Uniform** | 22.5 | 7.1 | 22.8 | 10.7 | 18.1 | 6.6 | 18.0 | 7.4 | 12.4 | 9.1 | 14.0 | -6.1 |
| **CPT Value** | 16.5 | 7.3 | 16.3 | 4.9 | 13.9 | 7.7 | 14.6 | -0.5 | 9.5 | 10.7 | 11.2 | -11.0 |
| **Wang (-0.75)** | 17.5 | 6.1 | 17.6 | 7.6 | 13.5 | 6.9 | 14.0 | 0.9 | 6.4 | 10.7 | 8.9 | -15.5 |
| **Wang (0.5)** | 23.3 | 6.3 | 23.3 | 13.1 | 18.5 | 6.0 | 18.8 | **8.2** | 12.9 | 9.0 | 14.4 | -5.6 |
| **Wang (0.75)** | 24.2 | 6.7 | 24.4 | 13.7 | 19.0 | 6.4 | 19.3 | 8.2 | **14.3** | 9.5 | **15.8** | **-4.4** |
| **Wang (1.0)** | 24.7 | 6.0 | 24.9 | 15.1 | **20.3** | 6.9 | **21.3** | 7.2 | 11.4 | 11.0 | 13.8 | -12.2 |
| **Wang (1.25)** | **25.4** | 6.1 | **25.4** | **15.9** | 17.3 | 6.7 | 17.8 | 5.6 | 12.7 | 10.4 | 14.8 | -8.3 |
| **Wang (1.50)** | 23.6 | 5.9 | 23.7 | 14.1 | 16.6 | 7.5 | 17.2 | 3.1 | 10.1 | 12.0 | 13.3 | -17.9 |
| **Wang (1.75)** | 23.4 | 6.2 | 23.5 | 13.5 | 12.5 | 8.0 | 13.0 | -0.6 | 7.3 | 13.0 | 11.1 | -22.8 |
| **PPO** | 19.0 | 6.4 | 18.9 | 9.1 | 15.8 | 6.0 | 15.8 | 5.8 | 8.3 | 9.5 | 9.5 | -9.8 |
| **PPO + Wang(0.75)** | 20.4 | 8.4 | 21.7 | 2.6 | 17.7 | 6.7 | 18.0 | 6.3 | 12.1 | 9.5 | 13.6 | -6.8 |

Table 1: Testing statistics for episode rewards achieved by agents trained over 10 million steps with different distributional objectives. $Q = 0.5$ is the median and $Q = 0.05$ refers to the location of the 0.05 quantile. Blue bold-face represents the best performance for a given environment; in all cases these occur for moderately cautious weightings ($\eta \in [0.75, 1.25]$).

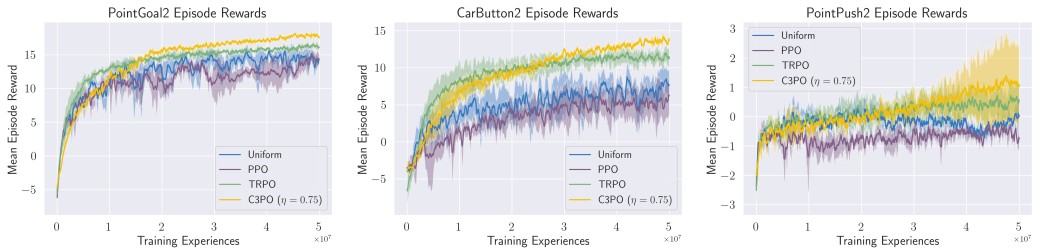

Figure 3: Average episode reward (including penalty) over training for different learning approaches in three different environments. C3PO with Wang ($\eta = 0.75$) weighting outperforms others.

average reward (including penalty) in five of the six environments and lowest average cost in five of the six environments. In addition, agents that used the cautious weightings tended to have more stable and repeatable training, as evidenced by the tight distribution of their learning curves. This tightness was found to reflect a lack of negative outlier episodes and potentially lower epistemic uncertainty throughout training. Finally, note that the use of a nonzero penalty for cost events resulted in significantly lower incurred costs than were observed with unconstrained agents trained without a penalty (Ray et al., 2019). PPO and TRPO were seen to reach similar cost levels without a penalty; these levels are indicated by red dashed lines in the cost figures.

Comparisons with versions of PPO and TRPO that use Lagrangian constraints (PPO-Lagrangian and TRPO-Lagrangian; Ray et al. (2019)) to match the cost level of C3PO are shown in Figure 6 and Appendix A.7 . We see that, given the same level of cost incurred per episode, agents trained using C3PO consistently achieve higher levels of reward than those trained with PPO-Lagrangian and TRPO-Lagrangian. As above, training is seen to be more stable and repeatable using our risk-sensitive method. Additional comparisons were generated with Constrained Policy Optimization (CPO; Achiam et al. (2017)), but are not shown in Figure 6 because they failed to maintain the cost levels of the other methods. For completeness, they are given in Appendix A.7.

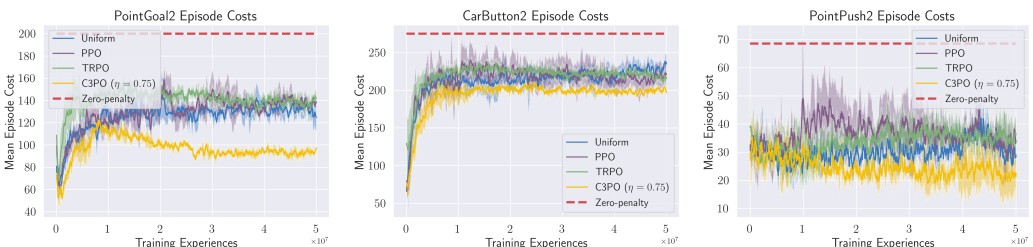

Figure 4: Average number of penalty events per episode (lower is better) over training for different learning approaches in three different environments. The horizontal lines reflect the cost levels reached by both PPO and TRPO training with zero penalty in Ray et al. (2019).

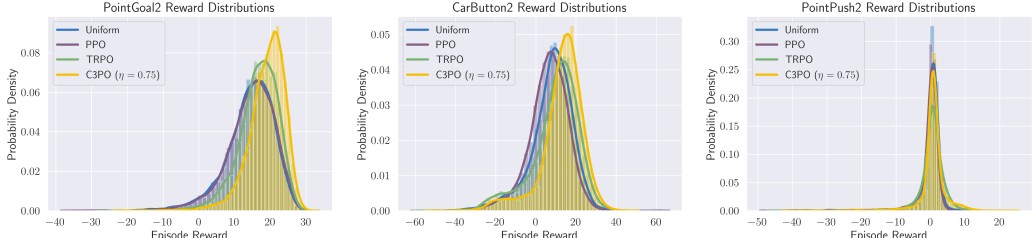

Figure 5: Testing reward distributions (including penalty; sampling turned off) for long training runs of three Safety Gym environments.

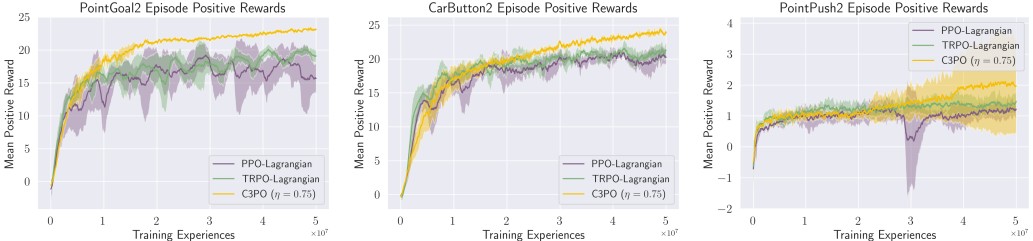

Figure 6: Comparison of positive contributions to episode reward during training for our approach (yellow) and Lagrangian methods configured to have the same cost level.

## 5 DISCUSSION

The analysis above allows for sample-based policy gradient estimates of a broad class of distributional objectives. Variance reduction measures were shown to enable efficient optimization based on these estimates (Appendix A.4). However, it was not seen to be the case that a given distributional objective could be most effectively optimized directly. Instead, the best results were generally obtained through moderate emphasis on improving negative training outcomes (Table 1).

To understand this behavior, consider the interplay of optimization and exploration in the training of cautious and aggressive agents. Cautious weightings continually emphasize the lower part of the outcome distribution, pushing that part of the distribution upward and adjusting behavior the most where it is most necessary. Once a part of the state space where the agent is deficient is adequately addressed, a different part of the state space takes its place. Policy entropy remains high because of the emphasis on problematic situations, ensuring adequate exploration. This trend continues with increasing $\eta$, until the point where the agent begins to ignore high quality training outcomes too much. Conversely, aggressive weightings continually emphasize the best outcomes in the distribution. When an already strong outcome is given increased attention, it is likely to stay at the top. Hence agents trained with aggressive weightings tend to become myopic, obsessing over a fraction of the state space while neglecting the rest of it. They tend to explore inadequately and ironically fail to attain better top-end performance than more cautious weightings.

Given the consistent performance gains observed using our method, we propose that it represents a useful option for improving the performance and stability of on-policy learners. This should be particularly true in the presence of a meaningful trade-off between positive and negative reward terms and when there is significant stochasticity. While our approach does add an additional hyperparameter – the shaping constant $\eta$ – one value for that hyperparameter was seen to provide gains across all environments tested. While our approach does not provide for a direct choice of cost limit as constrained methods do, it is simpler to implement and was consistently seen to be more performant for the cost level it reached. It is also likely possible to use cautious weightings in conjunction with constrained RL, though this has not yet been investigated.

## 6 CONCLUSIONS

In this work, we proposed a risk-sensitive learning algorithm based on a policy gradient estimate for a broad class of distributional objectives. When configured to emphasize improvement in scenarios where the agent performs poorly, we found our method to compare favorably with existing unconstrained and constrained on-policy learners.

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

## A  APPENDIX

### A.1  EVALUATION OF CROSS-TRAJECTORY TERMS IN POLICY GRADIENT

In this section, we first show that the cross-trajectory terms in our policy gradient estimate (6) have an expectation value of $0$. We then argue that their removal leads to a policy gradient estimate with reduced variance.

**Lemma 1.** *Cross-trajectory terms of the form $f(\tau_i, \mathbf{a}_{j,t}, \mathbf{s}_{j,t}) = u(r(\tau_i))\nabla_\theta \log \pi_\theta(\mathbf{a}_{j,t}|\mathbf{s}_{j,t})$, where $i \neq j$, do not contribute to the gradient estimate (6) in expectation.*

*Proof.* First, note that

$$E_{\tau_i \sim p_\theta(\tau), \tau_j \sim p_\theta(\tau)} f(\tau_i, \mathbf{a}_{j,t}, \mathbf{s}_{j,t}) = E_{\tau_i \sim p_\theta(\tau), \tau_j \sim p_\theta(\tau)} u(r(\tau_i))\nabla_\theta \log \pi_\theta(\mathbf{a}_{j,t}|\mathbf{s}_{j,t})$$

$$= E_{\tau_i \sim p_\theta(\tau)} \left[ u(r(\tau_i)) E_{\tau_j \sim p_\theta(\tau)} \left[ \nabla_\theta \log \pi_\theta(\mathbf{a}_{j,t}|\mathbf{s}_{j,t}) \Big| \tau_i \right] \right]$$

Then consider the innermost expectation:

$$
\begin{aligned}
E_{\tau_j \sim p_\theta(\tau)}\Big[\nabla_\theta \log \pi_\theta(\mathbf{a}_{j,t}|\mathbf{s}_{j,t})\Big|\tau_i\Big] &= \int_{\mathbf{s}_{j,t},\mathbf{a}_{j,t}} p(\mathbf{s}_{j,t},\mathbf{a}_{j,t}|\pi_\theta,\tau_i)\nabla_\theta \log \pi_\theta(\mathbf{a}_{j,t}|\mathbf{s}_{j,t})d\mathbf{a}_{j,t}d\mathbf{s}_{j,t} \\
&= \int_{\mathbf{s}_{j,t}} p(\mathbf{s}_{j,t}|\pi_\theta,\tau_i)\int_{\mathbf{a}_{j,t}} \pi_\theta(\mathbf{a}_{j,t}|\mathbf{s}_{j,t})\nabla_\theta \log \pi_\theta(\mathbf{a}_{j,t}|\mathbf{s}_{j,t})d\mathbf{a}_{j,t}d\mathbf{s}_{j,t} \\
&= \int_{\mathbf{s}_{j,t}} p(\mathbf{s}_{j,t}|\pi_\theta,\tau_i)\int_{\mathbf{a}_{j,t}} \nabla_\theta\pi_\theta(\mathbf{a}_{j,t}|\mathbf{s}_{j,t})d\mathbf{a}_{j,t}d\mathbf{s}_{j,t} \\
&= \int_{\mathbf{s}_{j,t}} p(\mathbf{s}_{j,t}|\pi_\theta,\tau_i)\nabla_\theta\int_{\mathbf{a}_{j,t}} \pi_\theta(\mathbf{a}_{j,t}|\mathbf{s}_{j,t})d\mathbf{a}_{j,t}d\mathbf{s}_{j,t} \\
&= \int_{\mathbf{s}_{j,t}} p(\mathbf{s}_{j,t}|\pi_\theta,\tau_i)(\nabla_\theta 1)d\mathbf{s}_{j,t} = 0.
\end{aligned}
$$

$\square$

To see why removal of cross-trajectory terms leads to reduced variance, consider that the full expression (6) may be written as the sum of terms of the form $f(\tau_i,\mathbf{a}_{j,t},\mathbf{s}_{j,t}) = u(r(\tau_i))\nabla_\theta \log \pi_\theta(\mathbf{a}_{j,t},\mathbf{s}_{j,t})$. Its variance is the sum of the total variance from terms where $i = j$, the total variance from terms where $i \neq j$, and a term proportional to the covariance of these two totals. However, because each term in the covariance contains at least one trajectory that differs from the rest, the above reasoning may be applied to argue that the covariance is 0. Hence, the removal of the cross-trajectory terms lowers the variance of the policy gradient estimate by the variance of the cross-trajectory terms.

## A.2 INTRODUCTION OF STATIC AND STATE-DEPENDENT BASELINES

**Lemma 2.** *A static baseline of the utility may be added to the policy gradient estimate (9) without introduction of bias.*

*Proof.* The additional term is 0 in expectation as

$$
\begin{aligned}
E_{\tau_i \sim p_\theta(\tau)}&\Big[b\Big(w'\Big(\frac{i}{n}\Big)+w'\Big(\frac{i-1}{n}\Big)\Big)\nabla_\theta \log \pi_\theta(\mathbf{a}_{i,t}|\mathbf{s}_{i,t})\Big] \\
&= b\Big(w'\Big(\frac{i}{n}\Big)+w'\Big(\frac{i-1}{n}\Big)\Big)\int_{\mathbf{s}_{i,t},\mathbf{a}_{i,t}} p(\mathbf{s}_{i,t},\mathbf{a}_{i,t}|\pi_\theta)\Big[\nabla_\theta \log \pi_\theta(\mathbf{a}_{i,t}|\mathbf{s}_{i,t})\Big]d\mathbf{a}_{i,t}d\mathbf{s}_{i,t} \\
&= b\Big(w'\Big(\frac{i}{n}\Big)+w'\Big(\frac{i-1}{n}\Big)\Big)\int_{\mathbf{s}_{i,t}} p(\mathbf{s}_{i,t}|\pi_\theta)\int_{\mathbf{a}_{i,t}} \pi_\theta(\mathbf{a}_{i,t}|\mathbf{s}_{i,t})\nabla_\theta \log \pi_\theta(\mathbf{a}_{i,t}|\mathbf{s}_{i,t})d\mathbf{a}_{i,t}d\mathbf{s}_{i,t} \\
&= b\Big(w'\Big(\frac{i}{n}\Big)+w'\Big(\frac{i-1}{n}\Big)\Big)\int_{\mathbf{s}_{i,t}} p(\mathbf{s}_{i,t}|\pi_\theta)\nabla_\theta\int_{\mathbf{a}_{i,t}} \pi_\theta(\mathbf{a}_{i,t}|\mathbf{s}_{i,t})d\mathbf{a}_{i,t}d\mathbf{s}_{i,t} \\
&= b\Big(w'\Big(\frac{i}{n}\Big)+w'\Big(\frac{i-1}{n}\Big)\Big)\int_{\mathbf{s}_{i,t}} p(\mathbf{s}_{i,t}|\pi_\theta)(\nabla_\theta 1)d\mathbf{s}_{i,t} = 0.
\end{aligned}
$$

The contribution of the weight terms $(w'(\frac{i}{n})+w'(\frac{i-1}{n}))$ may be pulled out of the integral between the first and second line because of its independence on both state and action. This term is fixed for a given trajectory by the rank of its reward amongst the rewards accumulated on all trajectories in the current batch.

$\square$

In our variance reduction experiments (Appendix A.4), the "Base" agent uses $b$ equal to the mean of full-episode utility in the current batch.

As described in Section 3.3, we may further adjust the policy gradient estimate through introduction of per-step utilities. In this case, we may justify the use of a state-dependent baseline through the following.

**Lemma 3.** *A state-dependent baseline $V_\phi(\mathbf{s}_{i,t})$ may be added to the policy gradient estimate (9) without introduction of bias, if per-step utilities are assumed.*

*Proof.* The additional term is 0 in expectation as

$$E_{\tau_i \sim p_\theta(\tau)}\left[\left(w'\left(\frac{i}{n}\right) + w'\left(\frac{i-1}{n}\right)\right)\nabla_\theta \log \pi_\theta(\mathbf{a}_{i,t}|\mathbf{s}_{i,t})V_\phi(\mathbf{s}_{i,t})\right]$$

$$= \left(w'\left(\frac{i}{n}\right) + w'\left(\frac{i-1}{n}\right)\right)\int_{\mathbf{s}_{i,t},\mathbf{a}_{i,t}} p(\mathbf{s}_{i,t},\mathbf{a}_{i,t}|\pi_\theta)V_\phi(\mathbf{s}_{i,t})\nabla_\theta \log \pi_\theta(\mathbf{a}_{i,t}|\mathbf{s}_{i,t})d\mathbf{a}_{i,t}d\mathbf{s}_{i,t}$$

$$= \left(w'\left(\frac{i}{n}\right) + w'\left(\frac{i-1}{n}\right)\right)\int_{\mathbf{s}_{i,t}} p(\mathbf{s}_{i,t}|\pi_\theta)V_\phi(\mathbf{s}_{i,t})\int_{\mathbf{a}_{i,t}} \pi_\theta(\mathbf{a}_{i,t}|\mathbf{s}_{i,t})\nabla_\theta \log \pi_\theta(\mathbf{a}_{i,t}|\mathbf{s}_{i,t})d\mathbf{a}_{i,t}d\mathbf{s}_{i,t}$$

$$= \left(w'\left(\frac{i}{n}\right) + w'\left(\frac{i-1}{n}\right)\right)\int_{\mathbf{s}_{i,t}} p(\mathbf{s}_{i,t}|\pi_\theta)V_\phi(\mathbf{s}_{i,t})\nabla_\theta \int_{\mathbf{a}_{i,t}} \pi_\theta(\mathbf{a}_{i,t}|\mathbf{s}_{i,t})d\mathbf{a}_{i,t}d\mathbf{s}_{i,t}$$

$$= \left(w'\left(\frac{i}{n}\right) + w'\left(\frac{i-1}{n}\right)\right)\int_{\mathbf{s}_{i,t}} p(\mathbf{s}_{i,t}|\pi_\theta)V_\phi(\mathbf{s}_{i,t})(\nabla_\theta 1)d\mathbf{s}_{i,t} = 0.$$

The rationale for pulling the $w'$ terms out of the integral is the same as in Lemma 2. □

Finally, we note that the ability to pull the contribution of the weight terms $(w'(\frac{i}{n}) + w'(\frac{i-1}{n}))$ to the front of Equation 11 allows us to formulate advantage estimates based on per-step utility. Bootstrap estimates of the value function $V_\phi(\mathbf{s}_{i,t})$ and Generalized Advantage Estimation as in Schulman et al. (2016) can be conducted exactly as they are in standard on-policy learning, if rewards are replaced by per-step utilities.

## A.3 ADDITIONAL INFORMATION ON SAFETY GYM

As mentioned in Section 4, we chose to evaluate our approach using the OpenAI Safety Gym (Ray et al., 2019). The choice was governed by our desire to test in conditions with clear cost-benefit trade-offs, significant stochasticity, adequate complexity, and available benchmarks. While our methods are not limited to particular task types or observation/action spaces, we found Safety Gym to be suitable for exploring their potential.

The six environments chosen were the most obstacle-rich of the publicly available environments that used the "Point" and "Car" robots. The Point robot is constrained to the 2D plane and has two control dimensions: one for moving forward/backward and one for turning. The Car robot also has two control dimensions, corresponding to independently actuated parallel wheels. It has a freely rotating wheel and, while it is not constrained to the 2D plane, typically remains in it. While we expect our results to extend to the remaining default robot, "Doggo", we did not experiment with it because of the order of magnitude longer training times it exhibited in Ray et al. (2019).

Several types of obstacles and tasks were present in the environments we evaluated. In all cases, the robot is given a fixed amount of time (1000 steps) to complete the prescribed task as many times as possible and is motivated by both sparse and dense reward contributions. In the "Goal" environments, the robot must navigate to a series of randomly-assigned goal positions, with a new target being assigned as soon as a goal is reached. In the "Button" environments, the robot must reach and press a sequence of goal buttons while avoiding other buttons. In the "Push" task, the robot must push a box to a series of goal positions. The set of obstacles are different for each task; among the three environments there are a total of five different constraint elements (hazards, vases, incorrect buttons, pillars, and gremlins), each with different dynamics. See Ray et al. (2019) for further details.

## A.4 EMPIRICAL PERFORMANCE OF VARIANCE REDUCTION MEASURES

To gauge the impact of the variance reduction techniques outlined in Section 3.3, we evaluated their performance in maximizing the value function of Cumulative Prospect Theory (Tversky & Kahneman, 1992). As mentioned in Section 3.1, this function has two integrals of the form (2):

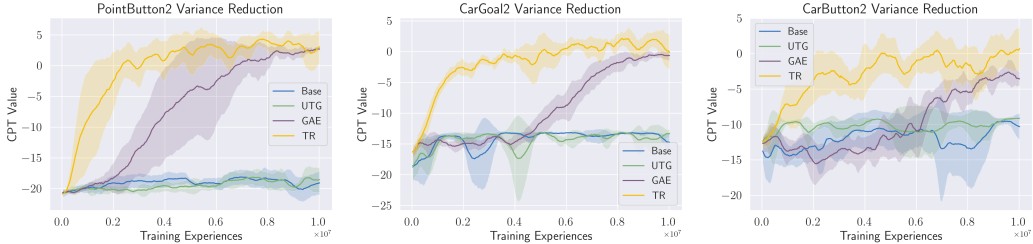

Figure 7: Impact of variance reduction measures on optimization of the CPT value function. Here "Base" refers to the risk-sensitive policy gradient estimate (10), "UTG" adds utility-to-go and a neural network baseline (11), "GAE" incorporates generalized advantage estimation, and "TR" implements trust regions via clipping. Shading represents the variation over five random seeds.

$$
\begin{aligned}
J(\theta) = &- \int_{-\infty}^{\infty} u^-(r(\tau)) \frac{d}{dr(\tau)} \bigg( w^- (P_\theta(r(\tau))) \bigg) dr(\tau) \\
&+ \int_{-\infty}^{\infty} u^+(r(\tau)) \frac{d}{dr(\tau)} \bigg( - w^+(1 - P_\theta(r(\tau))) \bigg) dr(\tau)
\end{aligned}
\tag{15}
$$

In Tversky & Kahneman (1992), the utility functions are computed relative to a reference point and reflect the tendency of humans to be more risk-averse in the presence of gains than in the presence of losses. The weight functions $\{w^+, w^-\}$ model our inclination to emphasize the best and worst possible outcomes in our decision-making.

More specifically, in these experiments we used the piecewise utility functions $u^+(r) = H(r - r_0)(r - r_0)^\sigma$ and $u^- = \lambda H(r_0 - r)(r_0 - r)^\sigma$ with static reference $r_0 = 10$, $\sigma = 0.88$, and $\lambda = 2.25$. The weight function $w(p) = \frac{p^\eta}{(p^\eta + (1-p)^\eta)^{\frac{1}{\eta}}}$ was used, where $\eta = 0.61$ for $r < r_0$ and $\eta = 0.69$ for $r \geq r_0$. Four methods were evaluated, incorporating progressive amounts of variance reduction:

- **Base**: Risk-sensitive policy gradient with a static baseline (10)

- **UTG**: Base with utility-to-go and a neural network baseline (11)

- **GAE**: UTG with generalized advantage estimation ((13) without clipping)

- **TR**: GAE with trust regions ((13) with clipping (14))

As shown in Figure 7, the incorporation of these techniques increased the sample efficiency of the CPT value optimization significantly. Consequently, we used the full complement (TR) in all other experiments.

## A.5 Differing Objectives: Additional Results

Below we include results for all environments for the experiments described in Section 4.1.

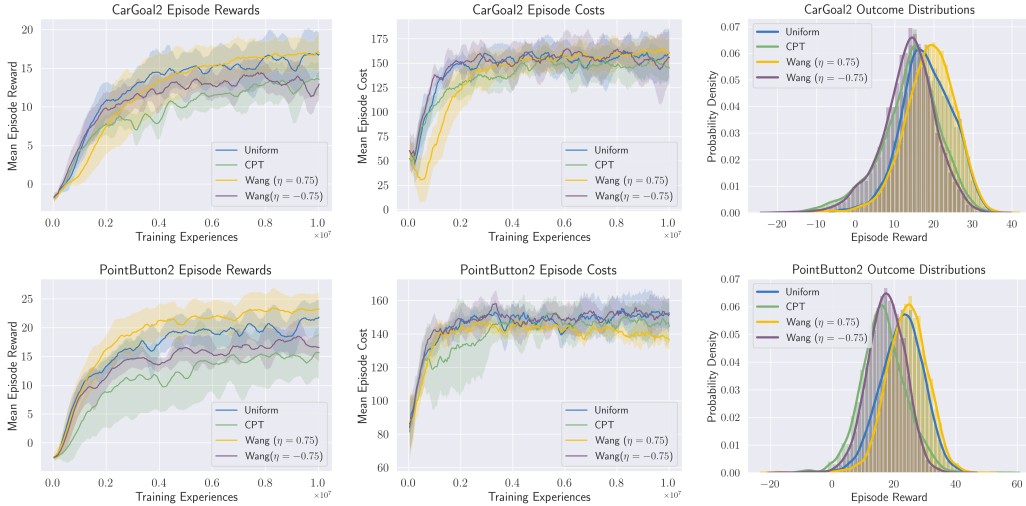

Figure 8: Impact of different distributional objectives in remaining two environments of initial trials. The shading reflects the standard deviation associated with running over 5 random seeds. Left: Net reward (positive reward minus penalty) throughout learning. Middle: Average number of cost events per episode during training (lower is better). Right: Agent outcome distribution in testing (i.e., with sampling turned off).

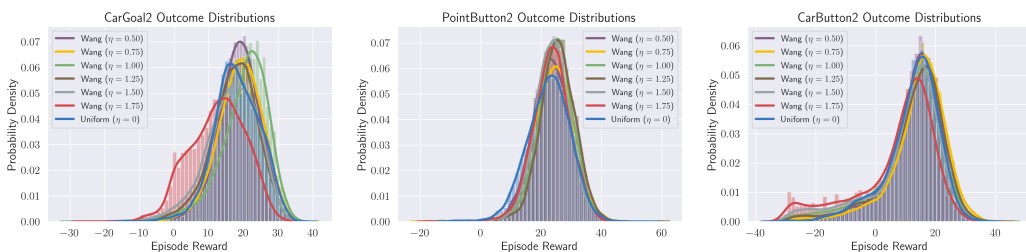

Figure 9: Agent outcome distributions across trials run over increasingly cautious ($\eta$ increasing) objectives. Distributions correspond to results shown in Table 1.

In addition, we note the trend of policy entropies with different distributional objectives. In general, more cautious weightings maintain higher entropy for longer than more aggressive weightings. Note that these plots represent an upper bound because they do not account for action clipping by the environment; see Ray et al. (2019) for details.

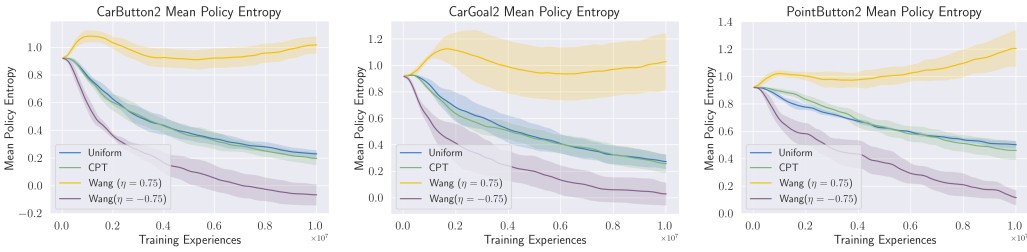

Figure 10: Policy entropy progression during training for three environments. Shading reflects the observed variation over 5 random seeds.

### A.6 ADDITIONAL COMPARISONS WITH UNCONSTRAINED METHODS

Below are plots of average episode reward and average number of episode cost events throughout training for the remainder of the environments on which we conducted long runs (Section 4.2). Also included are histograms of testing performance for those runs.

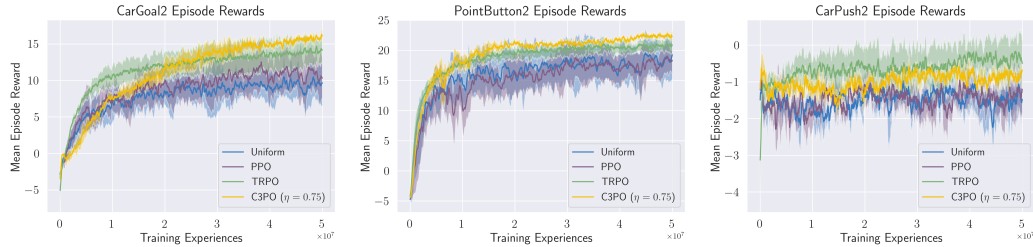

Figure 11: Average episode reward (including penalty) over training for different unconstrained learning approaches in remaining three environments.

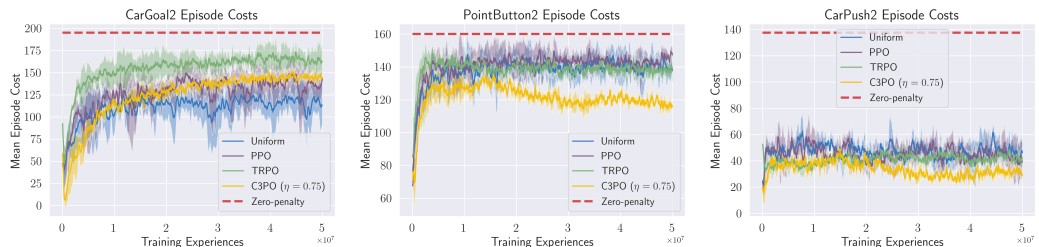

Figure 12: Average number of cost events per episode (lower is better) over training for different unconstrained learning approaches in remaining three environments. As above, the "zero-penalty" line refers to the level reached by PPO and TRPO trained with no penalty in the reward (Ray et al., 2019).

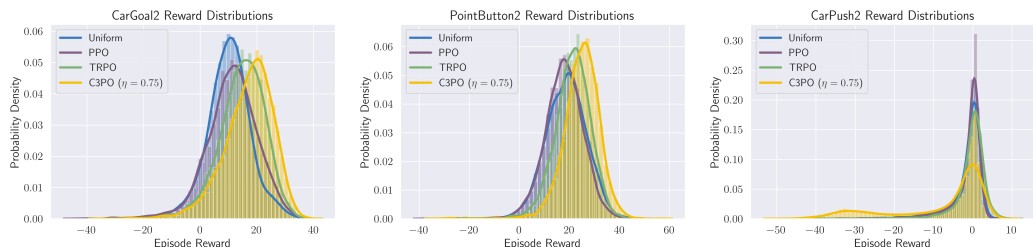

Figure 13: Testing reward distributions (including penalty; sampling turned off) for long training runs in the remaining three Safety Gym environments. In five of the six environments, C3PO with $\eta = 0.75$ provides tangible benefit. A smaller $\eta$ is likely required to improve performance on CarPush2.

### A.7 ADDITIONAL COMPARISONS WITH CONSTRAINED METHODS

As mentioned in Section 4.2, we compared the performance of C3PO with constrained methods by setting the cost limit of the constrained methods to match the cost level attained by C3PO. Here we provide

- the positive reward plots for the remaining three Safety Gym environments studied,
- the cost plots for each of the six environments, and

- all plots for Constrained Policy Optimization (CPO; Achiam et al. (2017)).

The intent of the cost plots of Figure 15 is to show rough consistency between the cost levels of our approach and Lagrangian methods configured to have the same cost limit. This is verified, but other trends should be noted. First, while the Lagrangian-based methods typically follow the cost constraint well, they cannot satisfy it in each batch. Second, our approach tends to have comparable or lower *cost rates* throughout training. This safe exploration metric, defined in Ray et al. (2019), refers to the average cost per episode over all of training up to a given point.

Results related to Constrained Policy Optimization (CPO; Achiam et al. (2017)) are included here but not in the main text because, consistent with (Ray et al., 2019), we were not able to configure CPO to respect the cost levels of the other constrained methods. Here we show the cost levels reached by CPO compared with C3PO (Figure 16) as well as a comparison of the average episode rewards of the two (Figure 17). For the latter, we employed the penalty scaling used by C3PO to enable a fair comparison.

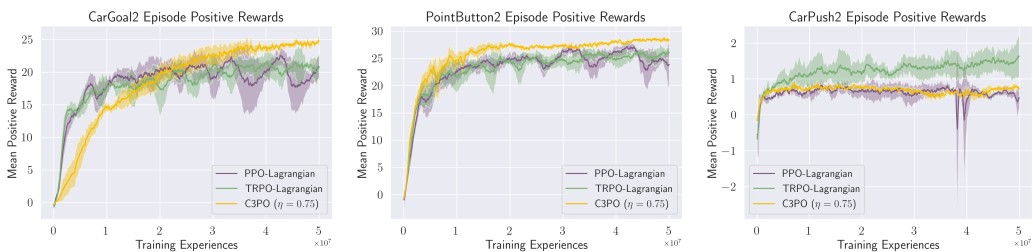

Figure 14: Comparison of positive contributions to episode reward during training for our approach (yellow) and Lagrangian methods configured to have the same cost level. The plots for the other three environments are shown in Figure 6.

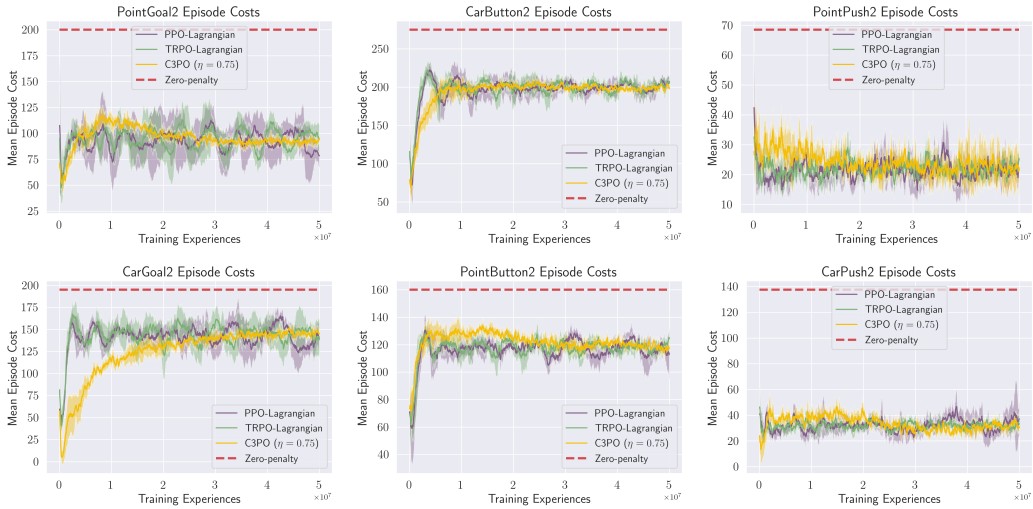

Figure 15: Comparison of cost incurred (lower is better) during training for our approach and Lagrangian methods configured to have the same cost level. As intended, cost levels are consistently matched between the methods. As above, the "zero-penalty" line refers to the level reached by PPO and TRPO trained with no penalty in the reward (Ray et al., 2019).

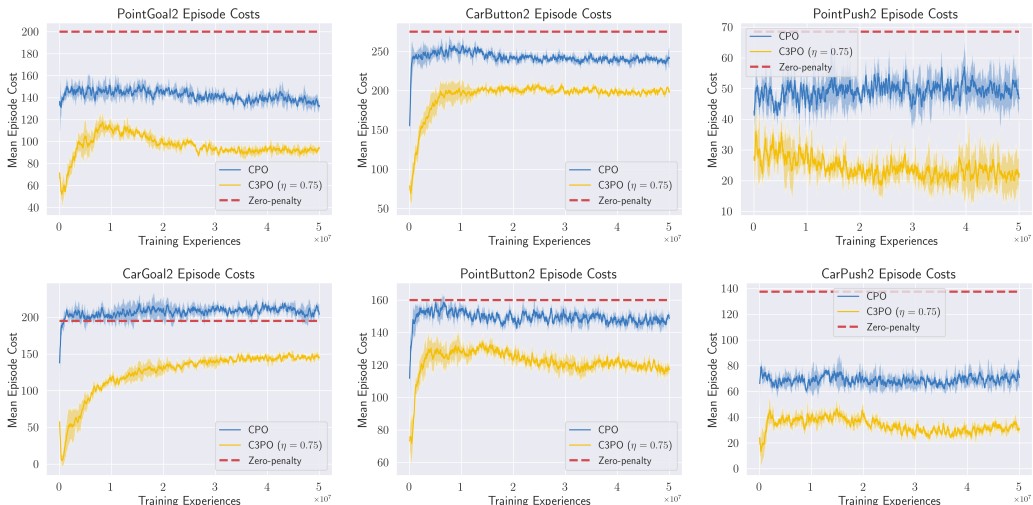

Figure 16: Comparison of cost incurred (lower is better) during training for our method and Constrained Policy Optimization (Achiam et al., 2017) configured to have a matching cost limit. Results are consistent with Ray et al. (2019). As above, the "zero-penalty" line refers to the level reached by unconstrained PPO and TRPO trained with no penalty in the reward (Ray et al., 2019).

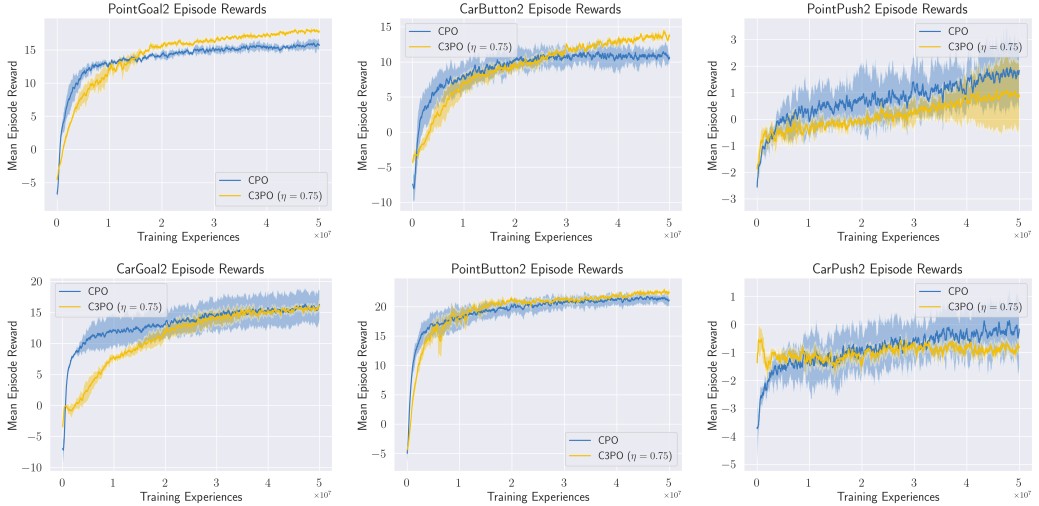

Figure 17: Comparison of average episode reward (including penalty) for C3PO and CPO.

## A.8 SUPPLEMENTARY MATERIALS

The code used to produce these results is included in our Supplementary Materials.

