# OpenReview forum: "A Risk-Sensitive Policy Gradient Method"
_ICLR.cc/2022/Conference — ICLR 2022 Submitted_

### Official Review · Reviewer_ugec · 2021-11-02

**Correctness:** 3
**Technical Novelty And Significance:** 2
**Empirical Novelty And Significance:** 2
**Recommendation:** 5
**Confidence:** 4

**Main Review:**

## Strenghts
The paper aims at solving an interesting problem, which is the optimization of a CPT inspired objective with a policy search approach. The authors derive a policy gradient for their objective and propose and estimator for it. They derive a PPO-like algorithm, which is then employed in the experiments.
The experimental analysis seems to be fair and sound: the experimental setting is described in a clear way, the experiments are repeated to take into account for the randomness in the weight initialization and the stochasticity of the performance is correctly represented in the plot with shaded areas, and by showing quantiles in the tables. The author included also the distribution of the returns, which helps understanding the impact of the risk-averse optimization.


## Weaknesses
While the author correctly described the state-of-the-art in the fields, some connections with related works are still not completely clear. For instance, it is not clear how much the proposed objective differs from distortion risk-measures (Wang, (1996), Dabney, (2018)). Furthermore, the author says that their estimation approach is similar to the one used in (LA Prashant, 2016), but it is not clear to which extent the developed techniques are different from the ones proposed there. It would help the reader to explicitly describe these connections with prior works, in order to ease the understanding of the authors contribution.
From a methodological viewpoint, the policy gradient derivation lacks a clear separation between the exact gradient formula and its estimator. Moreover, the authors do not show that expression (6) represents an unbiased estimate of the gradient, as they claim. Results in (LA Prashant, 2016) seems to suggest that the estimate would be instead only consistent, hence, asymptotically unbiased.
For what concerns the experimental analysis, the authors focus mainly on evaluating their results under the average reward objective. This is not fair in principle, since the proposed approach is actually optimizing a different objective. It would be interesting also to see how the algorithm performs w.r.t. the objective it is optimizing, including its learning curves. Finally authors suggest that the positive effect due to risk-averse optimization is similar to what happens with prioritized replay. However, while the latter technique gives an in higher weight to errors in value function estimation (high TD-errors), risk-averse optimization gives an higher weight to actual lower returns, thus, their objectives are substantally different.

**Summary Of The Paper:**

The article propose a policy gradient method for optimizing a CDF based criterion, inspired by CPT.
By varying the weighting function inside the objective, it is possible to change the risk-aversion of the agent.
The authors derives the policy gradient for the aforementioned objective and propose an estimation technique for it.
Then, they propose an algorithm which extends PPO for optimizing their objective. Empirical analysis is carried on to evaluate the approach on some modified Safety Gym environments, in which a fixed negative rewards corresponded to adverse events.
The authors evaluate different objectives obtained by employing a Wang weighting function, with different values of the parameter $\eta$. They show that optimizing a cautious (or risk-averse) objective allow to obtain better results in terms of average reward w.r.t. optimizing an aggressive (or risk-seeking) one. By further exploring the parameter space, the author demonstrate that some risk-averse values of the parameters allow to outperform also the risk-neutral version of PPO w.r.t. the average reward objective.

**Summary Of The Review:**

While the paper analyses an interesting problem and its experimental analysis is sound, it is not clear how novel its derivation is. Moreover, the author did not provide a clear picture on the policy gradient method they propose, for which further proofs should be included. The main claim of the authors, that some degree of risk-aversion may help risk-neutral optimization, is not novel from an empirical point of view, but it is still largely unexplained from the theoretical one, for which, unfortunately, no contribution is given in this work.
Due to a balance between the aforemetioned weaknesses and strenghts I suggest a weak reject.

---

> ### Author Response · Authors · 2021-11-17
> **Thank you for your review!**
>
> Thank you for your insightful review.  Addressing the weaknesses you identified, particularly surrounding the connection to prior work and the exposition of the algorithm, has been a useful exercise that we believe has allowed us to strengthen the work.  Our specific responses to your comments and accompanying changes made to the document are given below:
>
> -	We have clarified the relation to other notable risk measures in our edits.  In fact, the form we chose for the objective (Equation 2) is very general.  It can be used to optimize all of the measures evaluated in Dabney (2018), including those proposed by Wang (1996).
>
> -	We have revised the language surrounding the derivation of our policy gradient estimate in an effort to make it more precise and to more clearly delineate our contribution from that of Prashanth et al (2016).  The trajectory ordering scheme we use and its justification are from Prashanth et al, but that is the extent of the overlap.
>
> -	We have tightened the exposition of our policy gradient estimate, moving some of the algebra to the Appendix to allow more room for explanation throughout (including further proofs/justifications in the Appendix).  The ordering-based sampling scheme of our policy gradient is only asymptotically consistent, as you noted and as shown in the context of CPT value estimation in Prashanth et al.
>
> -	Regarding the presentation of experimental results, we certainly agree that in some sense comparing methods on objectives that they were not optimized for is not fair.  We made the decision to present results based on reward (looking at both the negative and positive components) for two reasons.  First, it provides a common ground among objectives that allows for comparison of the willingness of different agents to trade penalties for more positive reward.  Second, as discussed in Section 5, it was not seen to be the case that the best performance in a given metric came from optimizing that metric.  Hence, while we did plot curves of the learning of all objectives, we believe the included comparisons to be most informative.  One additional note here: on the recommendation of reviewer SaK7 we added comparisons with constrained methods.  Comparing levels of positive reward accumulated and penalty events incurred with these methods also seems most natural.
>
> -	We concede that the connection with Prioritized Experience Replay is not direct and certainly concur with your articulation of the difference between that and our method.  Our comment that they are “similar in spirit” was not meant to suggest that the mechanisms by which they impact learning are equivalent—instead it was meant to draw a parallel with another method that focuses on experiences where the agent is somehow deficient—but upon further review we agree that it’s too tenuous of a connection.  Accordingly, we have removed it from the text.
>
> -	While no theoretical justification was given for the ability of risk-aversion to help with risk-neutral optimization, our discussion in Section 5 does provide intuition about the underlying mechanism in our method.
>
> Thank you once again for your review; we would welcome further discussion!

---

### Official Review · Reviewer_VVvx · 2021-11-02

**Correctness:** 3
**Technical Novelty And Significance:** 3
**Empirical Novelty And Significance:** 2
**Recommendation:** 6
**Confidence:** 2

**Main Review:**

Strengths: The generalization to consider the entire CDF with arbitrary weightings appears to be a novel contribution to the literature. Furthermore, the authors also propose a simple to instantiate procedure for these weightings and a sample based algorithms that just sorts the trajectory outcomes.

Other comments:
* How robust are these results to the more common deep RL benchmarks outside of the safety gym?
* Once the utility is assumed to additively decompose across each state, action is this not equivalent to simply defining the utility as the reward?
* The variance reduction is basically the same as the usual policy gradient (and follow directly from the fact that the expectation of the gradient of log prob is 0).
* In Algorithm 1, where is $\hat{u}$ coming in? Is this just the sum over t of $u(s_{i,t})$? There seems to be some confusion over the dependence on both state and action or only state for the utility -- same issue between (14) and (15).
* A $\frac{1}{N}$ factor appears to be inconsistent across Eqs (11), (12) and (14)
* It's not clear how important the particular choice of -0.025 for the negative reward is for the results or how it was chosen. In particular, this seems like it would have a complex relationship with the $\eta$ weighting.
* Figure 5 captions says 6 environments, but there is only three in the figure.

**Summary Of The Paper:**

This paper considers a generalization of the policy gradient method to optimize for arbitrary utility functions with weightings that depend on the entire CDF (rather than the expected reward). This generalization has two aspects: (1) a utility function on top of the trajectory reward and (2) a weighting function for the CDF of the trajectory reward with respect to which the expectation is performed.  The paper derives an expression for the policy gradient and also generalizes the standard variance reduction baselines. Inspired by the PPO loss, the authors then propose a clipped version of the policy gradient calling it C3PO and evaluate this on some benchmarks from the OpenAI Safety Gym, where it is found that the conservative weightings can offer improvements over the standard formulation.

**Summary Of The Review:**

The main technical contribution is a novel generalization of the policy gradient expression to consider CDF weightings and a sample based algorithm that weights sorted trajectories to compute the gradient. The variance reduction extensions are fairly trivial. The empirical evaluations are limited to the open AI safety gym and are conducted on a PPO inspired version of the policy gradient result. These look promising, but the role of an arbitrarily chosen reward modification makes it hard to interpret how much the results depend on this  and how robust they are.

---

> ### Author Response · Authors · 2021-11-17
> **Thank you for your review!**
>
> Thank you for the careful review and for the attention to detail that went into it.  We have corrected the typos you caught, tried to address your comments regarding extensibility to other environments, and clarified the relationship between $\eta$ and penalty scaling.  Our responses to your comments and changes based on them are listed below:
>
> -	While we only tested our methods in Safety Gym, we do expect them to translate well beyond it.  There are no restrictions on the types of observation or action spaces our approach can be applied to, for instance.  The algorithm does assume that batches are comprised of full episodes, but this assumption can be waived if the episodes are replaced by windows of sufficient length to be considered “outcomes.”  This strategy matches human decision-making and could be used to implement the method on Atari, for instance (more detail in Section 3.4).  We chose Safety Gym for these initial experiments because it allowed us to easily look at tradeoffs in accumulating positive and avoiding negative rewards, in highly stochastic environments.
>
> -	Your question about the relationship between per-step utility and reward brings up a subtle point.  First of all, in our Safety Gym experiments, the utility and reward are the same in all experiments except those using the original Tversky-Kahneman CPT objective.  More generally, once the step-wise utility is computed, it can be treated the same way as a reward.  However, our formulation of the utility does allow for cases where one can’t immediately define the per-step utility from the reward.  For instance, consider the case where full-episode utility is computed differently based on whether a threshold full-episode reward is reached (e.g., the threshold could be the reference point in Tversky and Kahneman’s original CPT formulation).  Then one does actually have to backtrack through the episode (as described in Section 3.3) once it has concluded to get the step-wise utilities.
>
> -	We concur that our variance reduction measures largely track those of the usual policy gradient and rely on the same mathematical grounding.  The steps taken to remove terms corresponding to multiple trajectories are novel to our knowledge (since these terms originate from the incorporation of the CDF), but are built on similar principles.
>
> -	Thank you for catching our lack of precision regarding the introduction of $\hat{u}$ and the dependence of per-step utility on both state and action.  As was stated in Algorithm 1, $\hat{u}$ was the discounted utility to go; however it didn’t appear elsewhere and its definition reduced clarity.  Consequently, we have removed it.  We have also added the dependency of per-step utility on action back into Equation 13 (formerly 15) and Algorithm 1.
>
> -	Thank you for catching the missing factors of $1/N$ in our intermediate equations; this was a typo and we have corrected it.
>
> -	Your observation on the relationship between the penalty scale and $\eta$ is important and was probably not addressed sufficiently in our first draft.  Clearly, these are related hyperparameters that can be tuned.  Dialing up $\eta$ and/or the penalty factor will make the agent more cautious; however if you go too far in the combination you will hurt the ability to accumulate positive rewards.  In the environments tested, there was a range of $\eta$ that improved performance across environments, once the penalty scaling was reasonably chosen.  Hence a reasonable strategy for a new scenario would be to first pick a penalty level that gives acceptable initial cost levels, start with $\eta=0.75$, and adjust if necessary.  We did not find it necessary to tune $\eta$ to see benefits in either our comparisons with unconstrained methods or our (new) comparisons with constrained methods.
>
> -	Thank you for catching the typo in the Figure 5 caption.
>
> Thank you once again for your review; we would welcome further discussion!

---

### Official Review · Reviewer_apyr · 2021-11-02

**Correctness:** 2
**Technical Novelty And Significance:** 3
**Empirical Novelty And Significance:** 2
**Recommendation:** 5
**Confidence:** 4

**Main Review:**

STRENGTHS

The authors consider an interesting problem. I believe that risk awareness is important in RL. The utilization of a different risk measure, other than the one already explored, can be beneficial to obtain better risk awareness of the learning agent.

WEAKNESSES.

The paper is difficult to follow. Mathematical notation lacks precision (for example it is unclear whether the authors consider a finite horizon, or an episodic or a continuing problem in Equation 1). Most of all, concepts related to risk measures should be better explained. When one looks at the "CPT objective" at the beginning of page 3, it is unclear which roles play the utility function and the weighting. I do understand that probably the authors are very familiar with the underlying theory, but most people in our field (Rl) might be not. Would be nice to give an intuition.
Further, many passages seem unnecessary, or they are poorly explained.

Example: why do the authors bother giving us two different forms of $\nabla_\theta P_\theta$ in Equations 4 and 5? And why the same happen in 7? Is there any utility in showing these two different versions of the same quantity?

Also, it is not clear to me whether Equation 6 is an unbiased estimator of Equation 3. Can the authors give an intuitive explanation of why $\omega'(i/N)$, and can they show the unbiasedness of such estimator?

Section 3.3 starts with "unfortunately ... (6) suffers from high variance". Is this due to the log-likelihood derivative (as in classic REINFORCE) or there is an additional motivation caused by the new objective?

Derivation in (9) is known by most of the people in our field. The authors can just cite classic works on baseline subtraction.
Further, starting from Equation 9, and in many equations that follow, $\mathrm{d}$ is missing in the integrals.

I am not convinced that

$$
\left(\omega '\left(\frac{i}{n}\right) + \omega '\left(\frac{i-1}{n}\right)\right)
$$

can be pulled out of the expectation $\mathbb{E_{\tau_i}}$. Can the authors clarify the passages in 13?

Why the approach in (14) should minimize the variance? While subtracting the value function $V$ (or utility function, here), is a standard technique, remains unclear whether with the influence of

$$
\left(\omega'\left(\frac{i}{n}\right) + \omega'\left(\frac{i-1}{n}\right)\right)
$$

still makes $V$ a proper baseline subtraction.

Further, the usage of the utility function $u$ is unclear. In Equation 3, the utility takes as input a scalar function (the reward), i.e., $u(r(\tau))$. It is unclear how the per-step utility $u(\mathbf{s}, \mathbf{a})$ in Equation 14 is defined, and it is also unclear how $u(\mathbf{s})$ in Equation 15 is defined.

Can the authors explain the usage of the euclidean norm in Equation 15? The value function and the utility are scalars, why do we need that? Did the author mean to write something like

$$
\mathcal{L}(\phi) = \sum_{i,t} \mathbb{E}\left[V_\phi(\mathbf{s}_{i, t}) - \sum_{t'=t}^{T_i} u(\mathbf{s}_{i,t'})\right]?
$$

It would be nice to see the derivation of the value function and the advantage in this new setting of the risk measures, for example introducing a risk-aware Bellman equation. It is otherwise difficult to understand why the advantage "is computed with the standard GAE except with per-step utilities in places of rewards". The correctness of this approach might be trivial for the authors, but not for me.

In Section 4, could the authors clarify what ar the robots "Point" and "Car", and what are the tasks "Goal", "Push" and "Button"? It would be nice to know what are these tasks about, and the dimensionality of action and state spaces.

One thing that I think was really not clear, is what should I expect from the experiments. What should be the idea of using the CPT objective instead of other measures? How this should help?

I have seen that making the agent risk-aware can lead to better performances, as highlighted in your experiments, but shouldn't this effect appear also with other risk measures (like VaR or CVaR)? Could the author clarify this point? I think that a comparison with other risk measures would have been beneficial.

Further, uncertainty comes both as an aleatoric component (stochasticity of the environment) and epistemic (trust in my own model). Looks like that this work is focused on the aleatoric component (which is fine). Do the authors plan to extend their work also to the epistemic uncertainty?



**Summary Of The Paper:**

Risk objectives have long been investigated in reinforcement learning (RL). Most of the focus has been on classic risk measures, like exponential utility, value-at-risk (VaR), conditional value-at-risk (CVaR), leaving out, however, the cumulative prospect theory (CPT) developed by Tversky and Nobel Prize Kahneman in 1992, which has not yet been considered.

The advantage of CPT is to better model human decision-making, still allowing a wide class of risk measures, based on the utility $u$ and weighting $\omega$.

Hence, the authors consider a new risk-aware objective. Following some derivation, they compute a sample-based estimation of the gradient of this new objective w.r.t. the policy parameters.

The authors propose a PPO-like algorithm (called C3PO), which incorporates the new, risk-aware, gradient estimator.

They perform an empirical analysis on some tasks of "Safety Gym", showing that proper risk-awareness helps increase the performance of classic PPO.

**Summary Of The Review:**

STRENGHT:

The authors propose a new policy gradient algorithm based on a risk measure that has not been considered in RL so far. The considered risk-sensitivity measure comes from the cumulative prospect theory and aims both to mimic human decision making, and to generalizes a large class of other risk measures.

I believe that risk-aware RL is important, as intelligent beings rarely optimize for the "average" case, but they often act optimistically or pessimistically, usually giving higher weights to rare events.

WEAKNESSES:

The paper lacks a proper background, clarity, and precision. The mathematical notation should be improved, and while some passages could be skipped, since they are widely known by most of the audience, some passages instead remain obscure and need further clarification. I think that some passages are wrong, and I am waiting for the authors' response.

The main objective of the paper remains unclear. What is the benefit of utilizing this method in RL? Why are other risk measures (like VaR or CVaR) not enough?

The authors did not compare their risk measures with others, which I think was instead necessary.

UPDATE
---------

As the author improved the clarity of their submission and clarified some of my doubts (for example explaining the lack of comparison with VaR & CVaR), I am raising the score from 3 to 5.

---

> ### Author Response · Authors · 2021-11-17
> **Thank you for your review! (1 of 2)**
>
> Thank you for the thorough review and insightful comments.  In particular, they led us to tighten up the language surrounding the math and try to address big-picture takeaways more clearly.  We believe that addressing the weaknesses you pointed out has allowed us to produce a significantly stronger paper.  Our responses to the points you raised, as well as the changes made to address them, are the following:
>
> -	Regarding the clarity of the paper: we have edited passages that may have been opaque and moved some material to the appendices in order to include additional explanation for the interested reader.  We have also tried to remove anything not strictly necessary.  We have adjusted the explanation of the CPT objective and also underscored the generality of the chosen distributional objective (2).
>
> -	The two forms of $\nabla_\theta P_\theta$ given in Equations 4, 5 and 7 are necessary for Equation 8 (formerly 10).  We have added a comment where they are introduced to make sure they are not regarded as superfluous.
>
> -	We have cleaned up the language regarding the relation of Equation (6) to Equation (3)- thank you for pointing out our lack of precision here.  (6) is an asymptotically correct, sample-based estimate of the gradient in (3).
>
> -	We have clarified the intuition of the use of the derivative of the weight function.  Its presence in our formulation is a result of the application of the chain rule, and it tracks the use of $w’$ in Cumulative Prospect Theory (Tversky and Kahneman, 1992).
>
> -	Regarding the “high variance” of Equation (6): as you stated, the origin is the same as in classic REINFORCE.  In our case, it may be exacerbated by the different weighting of different experiences.  For instance, if the variance attributable to a heavily-weighted trajectory is high, one would expect the overall variance of (6) to be larger than that of the same data being evaluated by REINFORCE.
>
> -	We have added the “d’s” back to the integrals based on your recommendation.  We were aiming for brevity in dropping them but agree that putting them back is more clear/correct.  While we do believe the derivation formerly in (9) to be necessary in order to justify removal of cross-trajectory terms, we have moved it to Appendix A.1 to conserve space.
>
> -	Justification for taking the $w’$ out of the integrals in the former Equation 13 has been added; that derivation and the aforementioned justification are now in Appendix A.2.  The $(w’(i/n) + w’((i-1)/n))$ term is independent of state and action for a given trajectory, as it depends only on the rank of that trajectory’s reward relative to the other trajectories in the batch.  Hence it can be pulled out of the integrals in question.
>
> -	Regarding (14) minimizing variance: we did not claim that we had chosen $b$ to minimize variance but rather observed that one could in principle (by writing out the variance, differentiating with respect to $b$, setting equal to 0, and solving for $b$).  We have removed this statement to improve clarity.
>
> -	We have added justification for the use of a state-dependent baseline $V$ to Appendix A.2.
>
> -	The dependency of $u$ in (15) was a typo- thank you for catching it.  The utility $u$ is a function of the reward, which is in turn a function of state and action.  When defining a per-step utility via the given procedure, one can write $u$ as a function of state and action.
>
> -	Regarding the Euclidean norm in (15): we have changed that to improve clarity.  As you suggest, there was no reason to use a norm symbol there (or in Algorithm 1)- thank you for catching that.
>
> -	Regarding the derivation of the value function and advantage in this setting: since the weights can be “pulled out” of the expression, the formulation of GAE in terms of the value function follows Schulman et al (2016), except with $u$ in place of $r$.
>
> -	Based on your recommendation, we added Appendix A.3 to provide further information on Safety Gym.  It includes more specifics on the robots and tasks.
>
> -	Your question about what should be expected from the experiments is a good one.  While we attempted to articulate that at the beginning of the “Experiments” section of the initial draft, we have tried to clarify it in our revision.  In our initial experiments, we sought to establish that we could optimize different distributional objectives and that doing so led to differences in the outcome distributions.  We were hoping to find a distributional shaping that reduced the frequency of poor outcomes while not compromising much on the accumulation of positive reward, and we found that in the cautious weightings.  We then followed that observation with further experiments investigating the ability of these cautious weightings to both increase reward and reduce the frequency of cost events being incurred.
>
> Continued in next post!

---

> > ### Author Response · Authors · 2021-11-17
> > **Thank you for your review! (2 of 2)**
> >
> > -	Regarding comparison with other risk measures: our approach can be used to optimize many objectives, including VaR, CVaR and all of the measures in Dabney et al 2018a.  We have clarified that point in Section 3.1.  The three risk measures (beyond expected reward) chosen were motivated by a desire to have one approximate humans (CPT), one be aggressive in its decision-making, and one be cautious.  We did not use CVaR because we did not expect it to be competitive in terms of overall reward, given its neglect of good outcomes.  We certainly could add it though if critical.
> >
> > -	We appreciate the suggestion of framing our results in terms of aleatoric and epistemic uncertainty and have worked it into the text.  We concur that this study was more focused on the former; the choice of Safety Gym was largely based on its significant aleatoric uncertainty.  It is true, however, that in evaluating multiple random seeds of each agent, we are able to make some observations about epistemic uncertainty.  For instance, the histograms that we show are the result of running 1000 episodes over each of 5 random seeds.  These histograms then reflect a combination of aleatoric and epistemic uncertainty.  We also observed our method to typically have tighter distributions of learning curves (over multiple random seeds) than others, potentially reflecting smaller epistemic uncertainty.  One potential future direction related to epistemic uncertainty would be to see how the different agents performed when ensembled (both in groups trained using the same approach but different random seeds and in groups trained using different approaches).  Certainly an interesting direction.
> >
> > Thank you once again for your review; we would welcome further discussion!

---

> > > ### Comment · Reviewer_apyr · 2021-11-22
> > > **Answer**
> > >
> > > Dear authors,
> > >
> > > thank you for your careful answer.
> > > I think you have answered all my concerns. I read carefully both your answers and the new version of the paper.
> > >
> > > I will keep into consideration these clarifications and upgrade my grade accordingly.

---

> > > > ### Author Response · Authors · 2021-11-30
> > > > **Re: Answer**
> > > >
> > > > Thank you for being willing to reconsider- we look forward to your update!

---

> > > > ### Author Response · Authors · 2021-12-08
> > > > **Re: Answer**
> > > >
> > > > Thanks for your update.  Please let us know if you have particular concerns remaining that you would like to discuss!

---

### Official Review · Reviewer_SaK7 · 2021-11-03

**Correctness:** 3
**Technical Novelty And Significance:** 3
**Empirical Novelty And Significance:** 3
**Recommendation:** 6
**Confidence:** 3

**Main Review:**

### Strengths:

- The idea of using CPT in context to risk-sensitive RL seems promising. I also believe the approach is novel to the best of my knowledge.

- The paper is written clearly and with sufficient empirical rigour in the experiments conducted.

### Concerns:

- I think the clarity of some parts can be improved. A major component of the work is Eq 4, where the second transformation is not clear. Is it a property of the CDF or is it derived from the approach of Prashanth et al (2016.)? I think this is an integral component of the work and it should be clarified in more detail.

- The authors claim that the expression in Eqn 11 has a lower variance than expression in Eqn 6 but there is no accompanying proof. A formal explanation of this claim will help to strengthen the work.

- I also don’t understand why risk-sensitive DRL and constrained DRL approaches were excluded from the baselines. The comparison with PPO/TRPO demonstrates that the proposed approach can work better than the unconstrained methods. However, it is not clear what is the benefit of the proposed approach compared to the approaches in constrained RL literature. Some experiments with simple approaches like Lagrangian-PPO, CPO (which are both benchmarked and included with OpenAI gym) can help to make this distinction clear. Another possible method that can be included as a baseline is Cvar based objective (Chow et al 2015).



### Minor comments:

I think the notation for denoting CDF $P_{\theta}$ is a bit confusing as it gives the impression of parameterized transition model. I would encourage the authors to explore different notations for this quantity.


**Summary Of The Paper:**

The paper presents an alternate approach for distributional DRL via proposing an objective inspired from Cumulative Prospect Theory (CPT, Tversky and Kahneman, 1992). They use this distributional objective in conjunction with policy gradient methodology to propose a distribution policy gradient method for risk-sensitive RL. Under their approach, the distribution of the returns is optimized to maximize some chosen function of its CDF. They experiment with different such possible distributional objectives (risk profiles) on the OpenAI Safety Gym environments and show that their approach performs better than PPO.

**Summary Of The Review:**

I think the paper presents a really interesting approach to distributional DRL in context to risk-sensitive RL. Although there are some issues with the clarity, my biggest concern is that advantage of the proposed approach is not clear when compared to the existing literature on risk-sensitive and constrained RL. Although the paper could use more empirical strengthening, nevertheless, I think the will be useful to the community and hence recommend weak acceptance.

---

> ### Author Response · Authors · 2021-11-17
> **Thank you for your review!**
>
> Thank you for your careful review.  We went through your concerns and comments and believe that they helped to strengthen the paper.  In particular, your question about the relationship between our method and constrained approaches pushed us to do another set of experiments to more clearly situate our algorithm with respect to constrained approaches.  Our specific responses to the concerns you raised and changes based on them are the following.
>
> -	We did a significant refactor of the paper in an effort to address the concerns raised about overall clarity.  Specifically, we tightened our language and moved some derivations to the Appendix to provide space for further explanation of the material that raised concerns.  In this process, we did adjust the language around Equation 4 to better explain the second transformation and also made clear what is new from Prashanth et al (2016).  Prashanth et al contains justification for the ordering scheme we use for sampling (though in in the context of CPT value estimation), but is not used further.
>
> -	We have added justification for why Equation 9 (formerly 11) reduces variance from Equation 6; this is given in Appendix A.1.
>
> -	We ran a series of experiments to address your comment regarding the relation of our method to constrained approaches.  The results are shown in Figure 6 and Appendix A.7.  In these trials, we set the cost limit of the constrained methods to match the cost level reached by our approach.  While the ability of constrained methods to explicitly define desired cost levels ahead of time is a benefit, our method does consistently accumulate higher positive rewards for the same cost levels than constrained methods and is tunable through the $\eta$ parameter.  Finally, though we did not test it here, one could certainly optimize the CVaR objective using our approach (we have clarified this point in the text).  We chose not to include CVaR because we do not expect it to be competitive with other methods in terms of expected reward, given its neglect of good outcomes.
>
> -	Regarding the notation $P_\theta(r(\tau))$: we agree that it may be confusing for those familiar with model-based reinforcement learning.  After some deliberation we did decide to keep it though, as we like the relation to the trajectory probability $p_\theta(\tau)$, do not see a better alternative, and trust the reader to distinguish our method from a model-based approach.
>
> Thank you once again for your review; we would welcome further discussion!

---

### Author Response · Authors · 2021-11-23
**Updated Version Posted**

Thank you all again for your reviews- we found them to be constructive and believe that working through them has helped us to strengthen the paper significantly.  We posted a revision today (11/22) that includes a few minor corrections/improvements from the more significant update we posted on 11/17.  We would still welcome further discussion, as useful.

---

> ### Author Response · Authors · 2021-11-30
> **Please let us know if there are further issues to address!**
>
> Hi Reviewers,
>
> We see that the forum is still open for us, so please let us know if there are any additional points you would like us to address as you evaluate our responses and final draft.  Thanks again for your efforts!
>
> -- Authors

---

### Decision · Program_Chairs · 2022-01-20

**Decision:**

Reject

**Comment:**

This paper introduces a new approach for risk sensitive RL by using an objective that depends on the full distribution and can apply a weight to the resulting trajectory. The reviewers thought that focusing on more general and expressive objectives for RL is well motivated. However, they had a number of concerns of the current paper state, including its clarity in a number of sections and its relation to other work in risk-sensitive RL. The authors provided thoughtful responses but some concerns lingered around the prior concerns.